# Image Quality and Radiation Dose of Contrast-Enhanced Chest-CT Acquired on a Clinical Photon-Counting Detector CT vs. Second-Generation Dual-Source CT in an Oncologic Cohort: Preliminary Results

Florian Hagen [1,†], Lukas Walder [1,*,†], Jan Fritz [2], Ralf Gutjahr [3], Bernhard Schmidt [3], Sebastian Faby [3], Fabian Bamberg [4], Stefan Schoenberg [5], Konstantin Nikolaou [1] and Marius Horger [1]

1   Department of Diagnostic and Interventional Radiology, Eberhard-Karls-University, Hoppe-Seyler-Str. 3, 72070 Tübingen, Germany; florian.hagen@med.uni-tuebingen.de (F.H.); konstantin.nikolaou@med.uni-tuebingen.de (K.N.); marius.horger@med.uni-tuebingen.de (M.H.)
2   NYU Grossman School of Medicine, Department of Radiology, New York, NY 10016, USA; jan.fritz@nyulangone.org
3   Siemens Healthcare GmbH, 91052 Erlangen, Germany; ralf.gutjahr@siemens-healthineers.com (R.G.); bernhard.schmidt@siemens-healthineers.com (B.S.); sebastian.faby@siemens-healthineers.com (S.F.)
4   Department of Radiology, Albert-Ludwigs-University Freiburg, 79106 Freiburg, Germany; fabian.bamberg@uniklinik-freiburg.de
5   Department of Radiology, University of Mannheim, 68167 Mannheim, Germany; stefan.schoenberg@umm.de
*   Correspondence: lukas.walder@med.uni-tuebingen.de; Tel.: +49-07071-29-68622
†   These authors contributed equally to this work.

**Abstract:** Our aim was to compare the image quality and patient dose of contrast-enhanced oncologic chest-CT of a first-generation photon-counting detector (PCD-CT) and a second-generation dual-source dual-energy CT (DSCT). For this reason, one hundred consecutive oncologic patients (63 male, $65 \pm 11$ years, BMI: $16–42$ kg/m$^2$) were prospectively enrolled and evaluated. Clinically indicated contrast-enhanced chest-CT were obtained with PCD-CT and compared to previously obtained chest-DSCT in the same individuals. The median time interval between the scans was three months. The same contrast media protocol was used for both scans. PCD-CT was performed in QuantumPlus mode (obtaining full spectral information) at 120 kVp. DSCT was performed using 100 kV for Tube A and 140 kV for Tube B. "T3D" PCD-CT images were evaluated, which emulate conventional 120 keV polychromatic images. For DSCT, the convolution algorithm was set at I31f with class 1 iterative reconstruction, whereas comparable Br40 kernel and iterative reconstruction strengths (Q1 and Q3) were applied for PCD-CT. Two radiologists assessed image quality using a five-point Likert scale and performed measurements of vessels and lung parenchyma for signal-to-noise ratio (SNR), contrast-to-noise ratio (CNR), and in the case of pulmonary metastases tumor-to-lung parenchyma contrast ratio. PCD-CT $CNR_{vessel}$ was significantly higher than DSCT $CNR_{vessel}$ (all, $p < 0.05$). Readers rated image contrast of mediastinum, vessels, and lung parenchyma significantly higher in PCD-CT than DSCT images ($p < 0.001$). Q3 PCD-CT $CNR_{lung\_parenchyma}$ was significantly higher than DSCT $CNR_{lung\_parenchyma}$ and Q1 PCD-CT $CNR_{lung\_parenchyma}$ ($p < 0.01$). The tumor-to-lung parenchyma contrast ratio was significantly higher on PCD-CT than DSCT images ($0.08 \pm 0.04$ vs. $0.03 \pm 0.02$, $p < 0.001$). CTDI, DLP, SSDE mean values for PCD-CT and DSCT were $4.17 \pm 1.29$ mGy vs. $7.21 \pm 0.49$ mGy, $151.01 \pm 48.56$ mGy * cm vs. $288.64 \pm 31.17$ mGy * cm and $4.23 \pm 0.97$ vs. $7.48 \pm 1.09$, respectively. PCD-CT enables oncologic chest-CT with a significantly reduced dose while maintaining image quality similar to a second-generation DSCT for comparable protocol settings.

**Keywords:** photon-counting CT; dual-source dual-energy CT; radiation dose; image quality; chest-CT

## 1. Introduction

Computed tomography (CT) is a cornerstone of oncologic imaging due to its wide availability, comparably lower cost, excellent spatial resolution, and fast acquisition speed, enabling high throughput and contributing to patient satisfaction.

One major limitation of CT imaging is the accumulating radiation exposure in oncologic patients undergoing serial CT scans for treatment monitoring. Radiation dose-reducing techniques include using a lower tube current and tube voltage, automated exposure control, protocol tailoring to the clinical question, automated control of image quality, increased pitch, reduction in scan length, imaging filtering, improved detector efficiency, iterative reconstruction, and the use of individualized imaging protocols [1–5]. However, reducing radiation doses often reduces image quality, limiting its clinical utility.

With the advent of dual-source CT technology coupled with improved detector quality, dose reductions of up to 30% could be achieved already with third-generation dual-energy dual-source CT (DSCT) compared to second-generation DSCT [6]. Photon-counting detector CT (PCD-CT) technology is promising to further reduce radiation doses [7,8] as it converts individual X-ray photons directly into electronic signals proportional to their deposited energy, almost unaffected by electronic noise [9–12]. As PCDs do not have scintillators and septa, they can be fabricated with smaller elements compared to DSCT, thus significantly improving the spatial resolution [13]. Moreover, image quality can be further improved by manipulating the weighting of energy bins in the spectral data and also by using the decomposition of the material. For chest-CT applications, a great improvement in the detectability of both lesions with low-contrast compared to the adjacent lung parenchyma (e.g., partial solid nodules) as well as for high-contrast lesions has been reported by Si-Mohamed et al. [14]. Accordingly, we hypothesized that a first-generation PCD-CT should require less of a radiation dose than second-generation DSCT for standard contrast-enhanced chest-CT in oncologic patients while delivering comparable or improved image quality.

The purpose of our study was to compare the image quality and the patient dose of contrast-enhanced oncologic chest-CT of a first-generation PCD-CT and a second-generation DSCT using comparable exam protocol settings.

## 2. Materials and Methods

### 2.1. Subjects

Our institutional review board approved this prospective data evaluation, which was assigned the approval number 696/2021B01. Participants gave written informed consent. Between October 2021 and December 2021, a total of 100 consecutive oncologic patients who were referred for staging or treatment monitoring to our radiology department were enrolled (Table 1).

**Table 1.** Distribution of oncological disease among the participants.

| Oncological Diseases | Absolute Value (Relative Value in %) |
|---|---|
| NSCLC/SCLC | 12 (12%)/7 (7%) |
| Colorectal carcinoma | 17 (17%) |
| Gastrooesophageal carcinoma | 15 (15%) |
| Pancreatic carcinoma | 10 (10%) |
| Hepatobiliary cancer | 8 (8%) |
| Lymphoma | 7 (7%) |
| Others (ovarian carcinoma, thymic carcinoma, etc.) | 24 (24%) |

The same enrolled participants first underwent a standardized contrast-enhanced DSCT of the chest during either primary diagnosis and/or treatment monitoring, followed at the time by a PCD-CT examination using a comparable standardized protocol. All of the patients were enrolled consecutively, and none of them had to be excluded from the final evaluation. As differences both in image quality and radiation dose became evident

early in the course of the study, we decided to make a preliminary analysis of this cohort. The exclusion criteria included differences in the examinational protocols of PCD-CT and DSCT, contrast media concentrations, contrast media delay times, contrast agent volumes, and contrast media injection speed.

## 2.2. Dual-Source Chest-CT

DSCT studies were performed with the patients in a supine position using a 128-slice MDCT scanner (SOMATOM Definition Flash, Siemens Healthcare, Forchheim, Germany). The scan range comprised the entire chest, extending from the diaphragm to the thoracic inlet. We used the following examinational protocol: 100 kV for Tube A and 140 kV for Tube B, mean X-ray tube current of 100 mAs (automated tube current modulation called CareDose 4D was used), a slice thickness of 3 mm, focal spot 1.2, kernel I31f, iterative reconstruction ADMIRE1 (comparable to Q1 at PCD-CT), single collimation with a width of 0.6 mm, total collimation with a width of 57.6 mm, a table speed of 134.7 cm/s, a table feed/rotation of 38.6, a spiral pitch factor of 1.0, and a $512 \times 512$ matrix. The contrast medium protocol included intravenous administration of 1.2 mL/kg/body weight (IMERON 350 mg iodine/mL BRACCO Imaging, Germany) at a flow rate of 2 mL/s via the antecubital vein, followed by a 50 mL saline flush at 2.5 mL/s. The contrast material was administered using a dual-head pump injector (CT motion XD 8000, Ulrich Medical, Ulm, Germany). Arterial image acquisition started 31 s after intravenous administration and was the same for both scanners.

## 2.3. Photon-Counting Detector Chest-CT

PCD-CT was performed with the patients in the supine position using a first-generation dual-source CT scanner with quantum imaging (NAEOTOM Alpha, Siemens Healthineers, Forchheim, Germany) equipped with two photon-counting detectors. The polychromatic images were reconstructed for PCD-CT at 120 keV—so-called T3D— representing polychromatic information that can be considered comparable to a conventional polychromatic reconstruction on a DSCT scanner [15]. The following examinational protocol was used: 120 kV (automated tube current modulation—CareDose 4D), mean X-ray tube current 110 mAs, IQ level 60, a slice thickness of 3 mm, focal spot 0.8/1.2, kernel Br40f, iterative reconstruction factor Q1 and Q3, single collimation with a width of 0.4 mm, total collimation with a width of 57.6 mm, a table speed of 115.2 cm/s, a table feed/rotation of 57.6, and a spiral pitch factor of 1. The contrast agent protocol was standardized for all of the patients and was identical to the DSCT protocol.

## 2.4. Radiation Dose Quantification

In all of the patients, the volumetric CT dose index (CTDIvol) and dose length product (DLP) were obtained from the dose report, which was automatically stored in the picture archiving and communication system. The size-specific dose estimates (SSDE) were calculated based on each patient's effective diameter (transverse), as measured on the axial images [16]:

$$SSDE = (3.70 \times e - 0.0367 \times \text{effective diameter}) \times CTDIvol$$

Subsequently, the absolute values were compared between the two scanners.

## 2.5. Image Quality Quantification

Two radiologists with 1 (L.W.) and 2 (F.H.) years of experience in chest imaging performed the measurements. In each patient, round or oval ROIs were manually placed within the ascending thoracic aorta (ROI size, 80–120 mm$^2$), descending thoracic aorta (ROI size 80–120 mm$^2$), the pulmonary trunk (ROI size, 50–90 mm$^2$), as well as in the peripheral lung parenchyma (200–300 mm$^2$). For each region, three individual measurements were performed and averaged. The readers carefully avoided focal calcified aortic plaques or inhomogeneous areas of lung parenchymal attenuation, tumors, or areas of pulmonary

consolidation. Mean attenuation values were calculated by averaging the attenuation values of both radiologists. The signal-to-noise ratio (SNR) was calculated for each ROI as follows: SNR = (HUROI)/SDROI. Contrast-to-noise ratio (CNR) of the individual anatomic structures (lung and vessels) were calculated by manually placing ROIs with a size of 100–300 mm$^2$ in the patient's pectoral muscles and subcutaneous fat. Image noise was defined as the standard deviation (SD) of the subcutaneous fat (SDfat), and the organ-specific CNR was calculated as follows: CNR = (HUROI − HUmuscle)/SDfat.

In the patients presenting with pulmonary lesions (all of them metastatic in origin) ($n$ = 23), an additional ROI was placed in the main tumor manifestation (ROI size, 150–500 mm$^2$), and a ratio to the attenuation of lung parenchyma was calculated according to the formula:

$$\text{Tumor-to-lung parenchyma contrast ratio} = |(\text{ROItumor}/\text{ROIlung parenchyma})|$$

### 2.6. Subjective Image Quality

Two radiologists with 1 (L.W.) and 2 (F.H.) years of experience in chest imaging read all CT exams in a blinded, randomized, and independent fashion. Disagreements were resolved during a final consensus round ($n$ = 10 cases). The consensus reading consisted of a third joint measurement using a new ROI set together by both readers.

The images were randomly analyzed with freely-adjustable window settings. Subjective image contrast, image noise, and image sharpness were evaluated for the mediastinum, the lung, and the vessels by using a five-point Likert scale: five, excellent image quality; four, good image quality; three, fair but comprised image quality; two, poor image quality; one, non-diagnostic. Within the region of interest, both radiologists were free to choose their slice level. However, both readers had to complete a questionnaire suggesting the anatomical areas to be evaluated (e.g., upper lung lobes, the pulmonary arteries close to their first branching, ascending thoracic aorta 3 cm above the aortic valve level, descendent thoracic aorta about 3 cm above the diaphragms) and the ROI size ranges to be used.

### 2.7. Statistical Analysis

The data analysis was performed using IBM SPSS Statistics for Windows, Version 26.0 (IBM Corp., Armonk, NY, USA). The level of significance was set at $\alpha$ = 0.05. Continuous variables are provided as mean $\pm$ standard deviation (95% confidence interval) for normally distributed variables and median $\pm$ standard deviation (95% confidence interval) for non-normal data. Normal data distribution was assessed by applying the Shapiro–Wilk test. In the case of normal distribution, the variables of the two groups were compared according to the $t$-test for pairs. The Wilcoxon signed-rank paired test was used if data were not normally distributed. Comparison of DSCT with PCD-CT Q1 and Q3 iterative reconstruction strengths was performed with the Friedman test, followed by post-hoc Dunn–Bonferroni-tests with an alpha correction to analyze differences between the three subgroups.

## 3. Results

### 3.1. Patient Characteristics

A total of 100 participants (63 male, 65 $\pm$ 11 years) were included (Table 2). The mean time between both examinations was 3 $\pm$ 3.7 months.

### 3.2. Radiation Dose Quantification

The mean values of CTDI$_{vol}$ and DLP were 4.17 $\pm$ 1.29 mGy (1.98–9.38 mGy) and 151.01 $\pm$ 48.56 mGy * cm (64.8–312.0 mGy * cm) for the PCD-CT group, and 7.21 $\pm$ 0.49 mGy (6.59–11.42 mGy) and 288.64 $\pm$ 31.17 mGy * cm (233.2–479,6 mGy * cm) for the DSCT group, respectively ($p$ < 0.001 for both). The SSDE was 4.23 $\pm$ 0.97 (2.26–6.63) for the PCD-CT group and 7.48 $\pm$ 1.09 (3.81–10.47) for the DSCT group ($p$ < 0.001). The mean SSDE could be reduced by 43% compared to the previous DSCT examination (4.23 vs. 7.48).

**Table 2.** Patient characteristics.

|  | PCD-CT (Mean ± SD) | DSCT (Mean ± SD) | *p*-Value |
|---|---|---|---|
| Age (in [y]) | 65.02 ± 11.38 | 64.65 ± 11.14 | <0.001 * |
| Weight (in [kg]) | 72.34 ± 14.71 | 72.26 ± 14.85 | 0.142 * |
| Height (in [m]) | 1.72 ± 0.084 | 1.72 ± 0.086 | 0.620 * |
| BMI (in [$\frac{kg}{m^2}$]) | 24.43 ± 4.43 | 24.45 ± 4.53 | 0.304 * |
| Transverse diameter (in [cm]) | 34.67 ± 3.98 | 34.91 ± 4.17 | 0.141 ** |

* Wilcoxon signed rank paired test, ** paired *t*-test.

### 3.3. Image Quality Quantification

The objective analysis is displayed in Table 3. Q3 strength achieved the highest SNR for the vessels, lung parenchyma, subcutaneous fat, and chest muscle. Q1 strength had significantly lower SNR compared to DSCT. The CNR of Q1 and Q3 strengths achieved significantly higher values in the vessels compared to DSCT (18.02 ± 6.36 (Q1) and 22.48 (Q3) ± 8.07 vs. 11.98 ± 5.58 (DSCT), $p < 0.001$). The CNR of the lung parenchyma was highest for Q3 (PCD-CT), followed by ADMIRE1 (DSCT) and Q1 (PCD-CT) ($p < 0.001$).

**Table 3.** Comparison of the objective image quality between DSCT und PCD-CT.

|  | DSCT | PCD-CT | | *p*-Value | Corrected *p*-Value |
|---|---|---|---|---|---|
|  | ADMIRE 1 | Q1 | Q3 |  |  |
| ROI$_{ascending\_thoracic\_aorta}$ (Median ± SD) | 192.40 ± 42.28 | 282.05 ± 47.62 | 280.30 ± 45.34 | <0.001 | ** < 0.001 (Q3/DSCT) ** < 0.001 (Q1/DSCT) |
| ROI$_{descending\_thoracic\_aorta}$ (Median ± SD) | 182.90 ± 44.57 | 274.10 ± 45.85 | 273.90 ± 46.27 | <0.001 | ** < 0.001 (Q3/DSCT) ** < 0.001 (Q1/DSCT) |
| ROI$_{pulmonary\_trunk}$ (Median ± SD) | 208.25 ± 54.88 | 295.25 ± 77.71 | 294.65 ± 81.91 | <0.001 | ** < 0.001 (Q3/DSCT) ** < 0.001 (Q1/DSCT) |
| ROI$_{lung\_parenchyma}$ (Median ± SD) | −888.65 ± 29.22 | −888.65 ± 30.95 | −891.60 ± 31.25 | 0.403 |  |
| ROI$_{pectoralis\_muscle}$ (Median ± SD) | 63.60 ± 11.62 | 57.25 ± 8.00 | 57.10 ± 7.55 | 0.002 | ** 0.011 (Q3/DSCT) ** 0.010 (Q1/DSCT) |
| ROI$_{subcutaneous\_fat}$ (Median ± SD) | −96.65 ± 34.73 | −108.25 ± 18.47 | −108.55 ± 18.67 | <0.001 | ** < 0.001 (Q3/DSCT) ** < 0.001 (Q1/DSCT) |
| SNR$_{ascending\_thoracic\_aorta}$ (Mean ± SD) | 21.67 ± 5.98 | 19.44 ± 4.38 | 24.86 ± 5.36 | <0.001 * | ** Q3 > DSCT > Q1 ($p < 0.001$) |
| SNR$_{descending\_thoracic\_aorta}$ (Median ± SD) | 19.79 ± 6.77 | 19.01 ± 4.60 | 25.87 ± 6.35 | <0.001 * | ** 0.002 (Q3/DSCT) ** < 0.001 (Q1/DSCT) |
| SNR$_{pulmonary\_artery}$ (Median ± SD) | 20.81 ± 7.38 | 19.17 ± 5.38 | 25.45 ± 6.96 | <0.001 * | ** Q3 > DSCT > Q1 ($p < 0.001$) |
| SNR$_{chest\_muscle}$ (Median ± SD) | 5.62 ± 1.89 | 4.66 ± 3.57 | 6.02 ± 1.65 | <0.001 * | ** Q3 > DSCT > Q1 ($p < 0.001$) |
| SNR$_{tracheal\_air}$ (Median ± SD) | −110.72 ± 40.31 | −70.73 ± 25.48 | −84.14 ± 32.28 | <0.001 * | ** Q3 > DSCT > Q1 ($p < 0.001$) |
| SNR$_{subcutaneous\_fat}$ (Mean ± SD) | −8.30 ± 3.73 | −8.31 ± 2.52 | −10.59 ± 3.36 | <0.001 * | ** Q3 > DSCT > Q1 ($p < 0.001$) |
| SNR$_{lung\_parenchyma}$ (Median ± SD) | −78.61 ± 24.46 | −70.21 ± 20.51 | −85.15 ± 26.63 | <0.001 | ** < 0.001 (Q1/DSCT) ** < 0.001 (Q3/DSCT) |
| CNR$_{vessel}$ (Median ± SD) | 11.98 ± 5.58 | 18.02 ± 6.36 | 22.48 ± 8.07 | <0.001 * | ** Q3 > Q1 > DSCT ($p < 0.001$) |
| CNR$_{lung\_parenchyma}$ (Median ± SD) | −85.08 ± 22.61 | −75.00 ± 16.68 | −93.90 ± 26.44 | <0.001 * | ** Q3 > DSCT > Q1 ($p < 0.001$) |
| Tumor-to-lung parenchyma contrast ratio (Median ± SD) | 0.03 ± 0.02 | 0.08 ± 0.04 | 0.08 ± 0.04 | <0.001 * | ** Q3 > DSCT ($p < 0.001$) ** Q1 > DSCT ($p < 0.001$) |

* Friedman-test, ** Post-Hoc Dunn–Bonferroni-Tests with corrected alpha.

Q1 and Q3 strength resulted in a significantly higher tumor-to-lung parenchyma contrast ratio compared to the corresponding DSCT data sets. The maximum tumor size

was significantly larger at the time of PCD acquisition ($12.4 \pm 8.96$ cm vs. $11.00 \pm 8.63$ cm) (Table 3 and Figures 1 and 2).

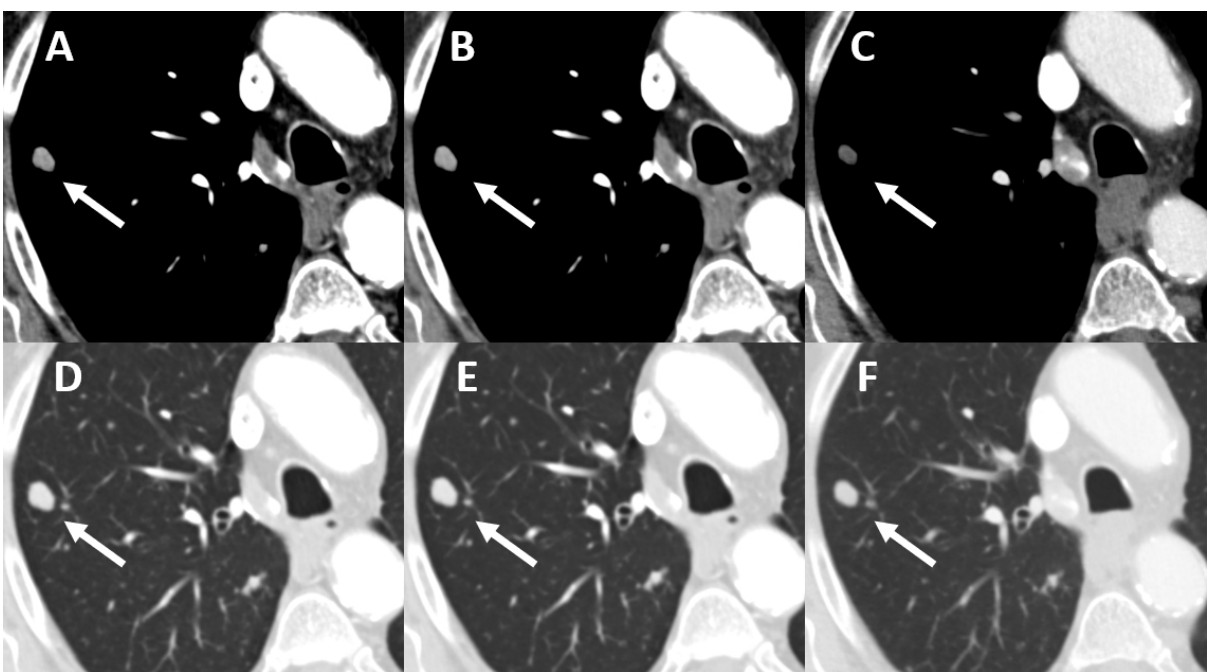

**Figure 1.** Pulmonary metastasis (arrows) of a colorectal carcinoma in a 75-year-old man. Tumor size: (**C**,**F**) 9 mm, (**A**,**B**,**D**,**E**) 10.5 mm, timespan between the examinations: 3 months.

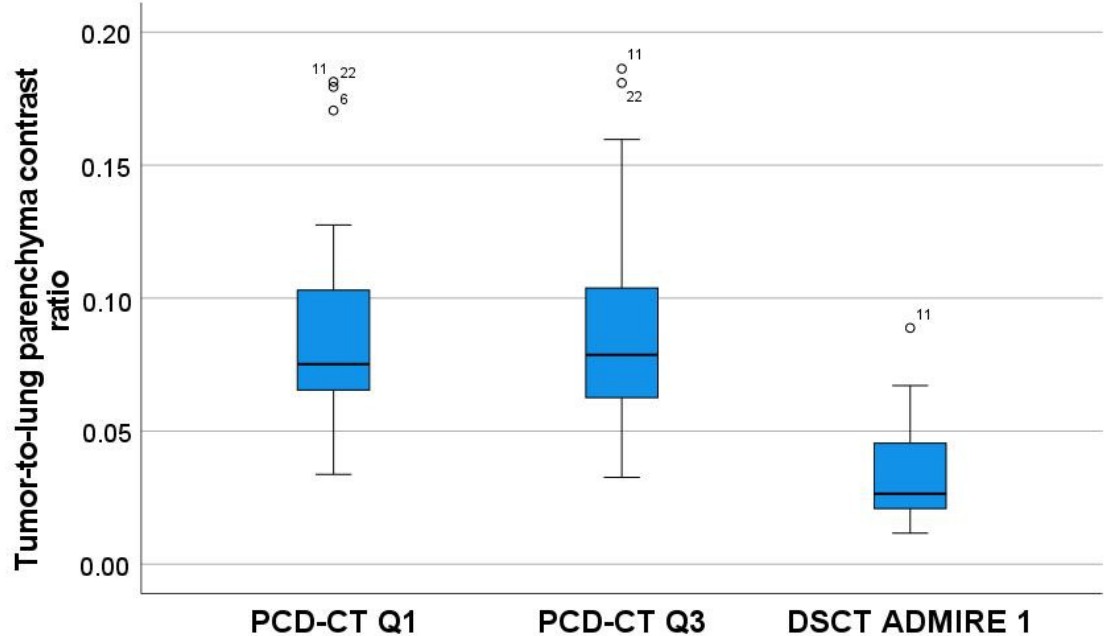

**Figure 2.** Boxplots representing the ratios of tumor-to-lung parenchyma contrast calculated for the two scanners, including the two Quantum strengths (Q1 and Q3) for the PCD.

A-C: Window width: 342 HU, Window level: 56 HU, D-E: Window width: 1500 HU, Window level: $-500$ HU, C and F: ADMIRE1 iterative reconstructed image data set, A and D: Q1 iterative reconstructed image data set, B and E: Q3 iterative reconstructed image data set.

### 3.4. Subjective Image Quality

After the first reading session, 10 image data sets (four male and six female patients) were separately rated by both radiologists. A common consent was reached after the final reading session. The median rating of overall image quality, noise, and contrast was four (good) for PCD-CT and DSCT. Nonetheless, especially the image contrast of the mediastinum, vessels, and lung parenchyma were rated significantly higher with PCD-CT compared to DSCT ($p < 0.001$) (see Figures 3–5).

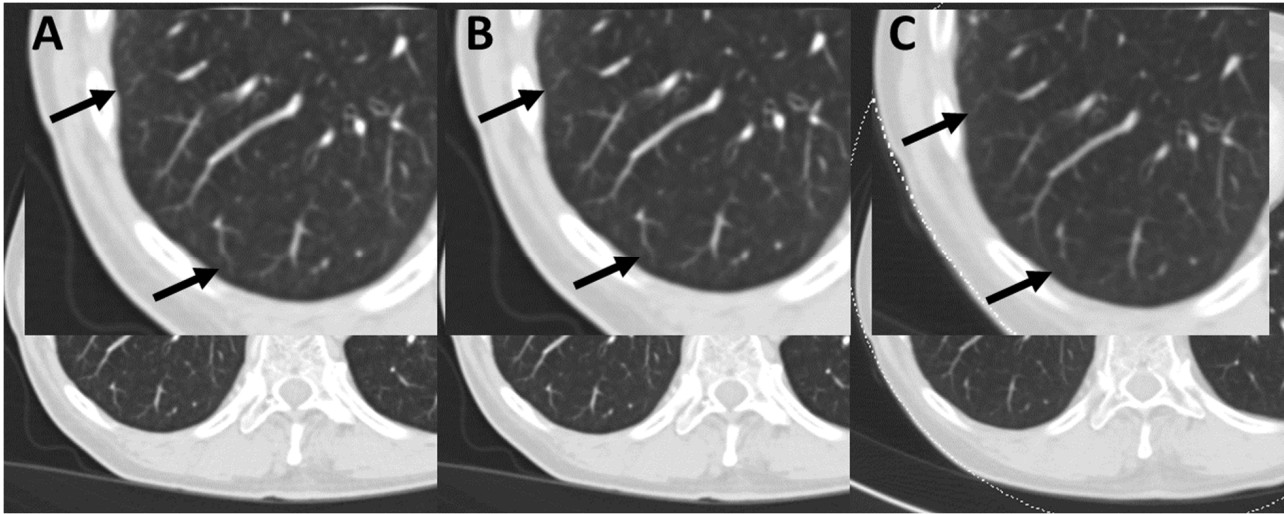

**Figure 3.** Delineation of lung vessel contours in a 59-year-old man. Window width: 1000 HU, Window level: −590 HU, (**A**) Q1 iterative reconstructed image data set, (**B**) Q3 iterative reconstructed image data set, (**C**) ADMIRE1 iterative reconstructed image data set. Arrows: bifurcation of thin vessels.

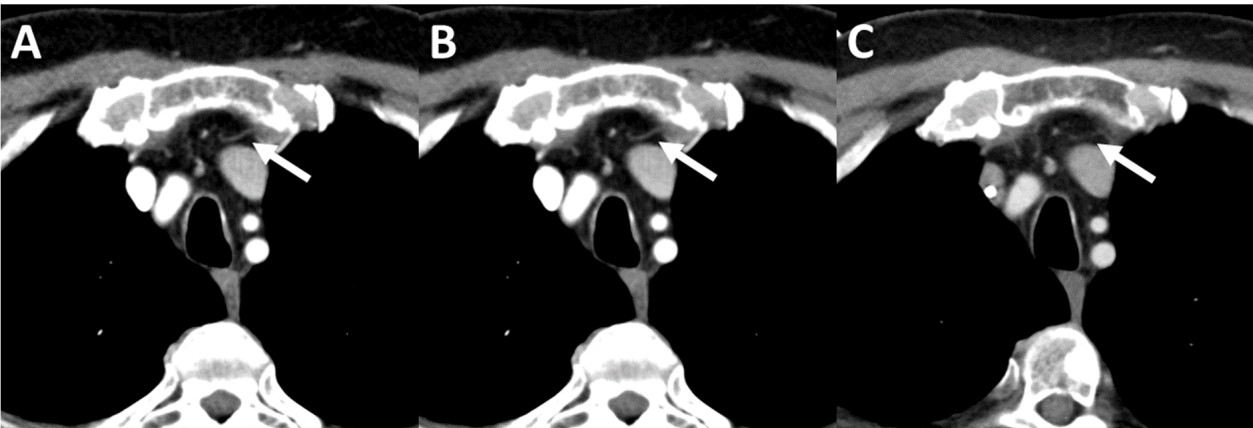

**Figure 4.** Delineation of mediastinal vessel contours in a 63-year-old man. Window width: 342 HU, window level: 56 HU (**A**) Q1 iterative reconstructed image data set, (**B**) Q3 iterative reconstructed image data set, (**C**) ADMIRE1 iterative reconstructed image data set. Arrows: thin mediastinal vessel.

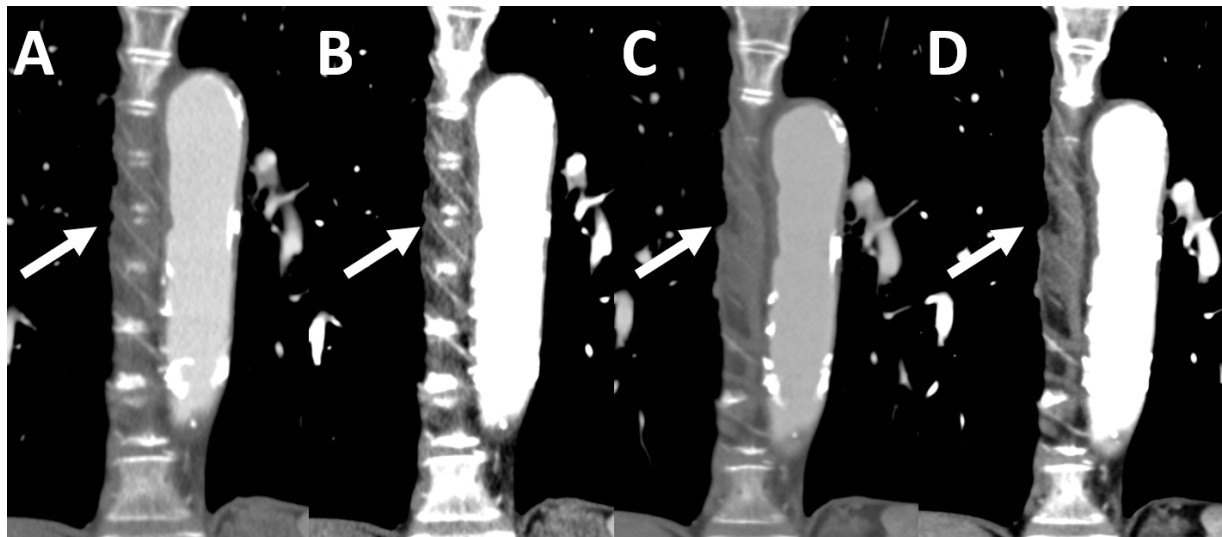

**Figure 5.** Coronal reformats of a contrast-enhanced chest-CT in a 76-year-old female patient. Note the different quality in delineation of intercostal vessels. (**A**,**C**): Window width: 877 HU, Window level: 108 HU, (**B**,**D**): Window width: 342 HU, Window level: 56 HU, (**A**,**B**) Q1 iterative reconstructed image data set, (**C**,**D**) ADMIRE1 iterative reconstructed image data set. Arrows: thin intercostal vessels.

The Q1 strength of the PCD-CT was associated with a higher noise level in the mediastinum, vessels, or lung parenchyma compared to the equivalent DSCT images (Table 4). Lung parenchyma sharpness was significantly higher in Q3 level reconstructed PCD-CT images than the DSCT images ($p = 0.027$).

**Table 4.** Comparison of the subjective image quality between the DSDECT and PCD-CT (Q1 and Q3).

| | | DSCT | PCD-CT | | *p*-Value | Corrected *p*-Value |
|---|---|---|---|---|---|---|
| | | Admire 1 | Q1 | Q3 | | |
| Mediastinum | Image Noise *Median (Min-Max)* | 4 (3–5) | 3 (2–4) | 3 (2–4) | <0.001 * | ** DSCT > Q1 ($p$ < 0.001) ** DSCT > Q3 ($p$ = 0.014) |
| | Image Contrast *Median (Min-Max)* | 3 (2–4) | 4 (2–4) | 4 (2–4) | <0.001 * | ** Q1 > DSCT ($p$ < 0.001) ** Q3 > DSCT ($p$ < 0.001) |
| | Image Sharpness *Median (Min-Max)* | 4 (2–5) | 3 (1–4) | 3 (2–5) | <0.001 * | ** DSCT > Q1 ($p$ = 0.003) ** DSCT > Q3 ($p$ = 0.024) |
| Vessels | Image Noise *Median (Min-Max)* | 4 (3–5) | 4 (3–5) | 4 (3–5) | <0.001 * | No differences between the groups |
| | Image Contrast *Median (Min-Max)* | 4 (2–5) | 5 (3–5) | 5 (3–5) | <0.001 * | ** Q1 > DSCT ($p$ < 0.001) ** Q3 > DSCT ($p$ < 0.001) |
| | Image Sharpness *Median (Min-Max)* | 4 (3–5) | 4 (3–5) | 4 (2–5) | 0.662 * | No differences between the groups |
| Lung parenchyma | Image Noise *Median (Min-Max)* | 4 (3–4) | 3 (2–4) | 4 (3–5) | <0.001 * | ** Q3 > Q1 ($p$ < 0.001) ** DSCT > Q1 ($p$ < 0.001) |
| | Image Contrast *Median (Min-Max)* | 4 (3–4) | 4 (3–5) | 4 (3–5) | <0.001 * | ** Q3 > DSCT ($p$ = 0.027) |
| | Image Sharpness *Median (Min-Max)* | 4 (3–5) | 4 (3–5) | 4 (3–5) | 0.004 * | No differences between the groups |

* Friedman-test, ** Post-Hoc Dunn–Bonferroni-Tests with corrected alpha.

## 4. Discussion

Our results show that the radiation dose for contrast-enhanced chest-CT can be significantly reduced with PCD-CT compared to second-generation DSCT, thereby yielding similar image quality. Based on SSDE, PCD-CT enabled a 43% dose reduction. Concomitantly, reader-based image quality assessment yielded comparable results of PCD-CT and second-generation DSCT with potential for further improvement with increasing Q

strength. There was no significant difference in body weight between DSCT and PCD-CT to suggest patient-based confounding factors. Quantitative image analysis showed the best SNR for the vessels, lung parenchyma, and subcutaneous fat for PCD-CT when using Q3, whereas Q1 was inferior to ADMIRE1 strength when applied to DSCT data. Combining vessel measurements of the aorta and pulmonary vessels resulted in a significantly higher CNR of PCD-CT for both Q1 and Q3 over DSCT.

Similarly, the visual reader assessment of image contrast and noise were best for PCD-CT in all tissues, but the differences to DSCT proved not significant. Moreover, the mediastinal noise levels were significantly lower for PCD-CT Q1 and Q3 strengths compared to DSCT. The image contrast was better for PCD-CT, even for the Q1 strength, than DSCT. For lung parenchyma, the image contrast and quality were significantly better for both Q1 and Q3 strengths. Finally, the tumor-to-lung parenchyma contrast ratio proved superior for the PCD over the EID.

The image quality of chest CT depends on many variables, which can be patient-dependent (BMI, compliance, size and density of organs and tumors), protocol-related (tube current, tube voltage, kernel, pitch, iterative reconstruction, individualization of scanning protocols depending on the clinical question, section thickness, section spacing, scan length), and hardware-dependent (beam-shaping filter and improved detector technology) [11,12].

In practice, CT image quality should comply with the clinical requirements. While reduced image quality may be sufficient for chest-CT screening, oncologic CT often requires high-diagnostic image quality to ensure the accurate detection of even subtle abnormalities. Despite previous reports recommending low-energy or even ultra-low-dose energy protocols for diagnosing pneumonia, lung nodules, and fibrosis [17,18], a meta-analysis evaluating the accuracy of low-dose-CT found widely varying accuracies that prevented deriving unified performance data [19]. Some reports indicated adequate detection accuracy of honeycombing and bronchiectasis with low-dose CT, as well as the adequacy of ultra-low-dose-CT for diagnosing pneumothorax, consolidations, and ground-glass opacities. Others have stated that low-dose chest-CT protocols are potentially beneficial in certain clinical settings but not in obese patients or with atypical interstitial lung diseases [20]. In contrast, oncologic chest-CT has a wide spectrum of abnormalities and complex clinical questions, including tumor monitoring, the detection of new metastases, the assessment of potential drug toxicity-induced pulmonary complications, and paraneoplastic coagulation disorders with aortic or pulmonary vascular thrombosis for which adequate image quality is a requisite for accurate diagnoses.

With dose reduction remaining a major goal in clinical practice, PCD-CT may bridge the gap between dose reduction and image quality comparable to the high image quality of conventional DSCT. The main difference between conventional energy-integrating detector (EID) CT and PCD-CT is that the former uses indirect conversion technology, with a layer of scintillators converting X-ray photons into visible light, which are consequently detected by a photodiode and converted into electronic signals, whereas the latter directedly converses X-ray photons into electron-hole pairs by using a semiconductor detector material with a better electron yield. Contrary to the conventional energy-integrating detector CT, which integrates the energy levels of all detected photons, PCDs count the number of individual photons exceeding a specified energy level. Therefore, the electronic noise is usually negligible for protocols used in average-sized patients [20]. Electronic noise is usually detected as a low-amplitude signal. PCD-CT excludes electron noise by setting a slightly higher low energy threshold than the energy level associated with the electronic noise signal amplitude. Based on the physical principles of PCD-CT, noise reduction, increased spatial resolution, and dose reduction have been anticipated [21–24]. This explains our results showing a clear trend towards higher CNR and lower noise levels. Regarding artifacts, PCD-CT has the potential to reduce blooming by means of improved spatial resolution and material decomposition [21].

PCD-CT increased iodine contrast with similar image noise compared to energy-integrating detectors, thereby realizing a dose reduction of 32% [25]. Dose reductions in a

similar range were also described for chest, head, and neck applications [26,27]. The results of our oncologic chest CT are in keeping with the published data.

Our study has limitations. We did not intend to adjust the two protocols to comparable energies but to image quality. Based on the major technical differences between the DSCT and PCD-CT detector technologies, protocol parameters and applied energies may not be translatable in an identical fashion. Only one iterative reconstruction strength (1) was available for the second-generation DSCT, which limited comparability to PCD-CT. Some major aspects related to the superiority of PCD over EID in the chest diagnosis in terms of lesion detectability and improved spatial resolution for accurate display and interpretation of, e.g., interstitial lung diseases have not been addressed in this study.

## 5. Conclusions

In conclusion, PCD-CT enables oncologic chest-CT with a significantly reduced dose while retaining image quality similar to a second-generation DSCT.

**Author Contributions:** Conceptualization, M.H. and F.H.; Methodology, M.H. and K.N.; Formal Analysis, L.W. and F.H.; Investigation, M.H., F.H. and L.W.; Data Curation, F.H.; Writing—Original Draft Preparation, M.H., J.F., F.H. and L.W.; Writing—Review & Editing, M.H., J.F., B.S., S.F., R.G., F.B. and S.S. All authors have read and agreed to the published version of the manuscript.

**Funding:** This project is funded by the Baden-Württemberg Ministry of Economic Affairs, Labor and Tourism as part of the "Forum Gesundheitsstandort Baden-Württemberg, number: 35-4223.10/20.

**Institutional Review Board Statement:** The study was conducted in accordance with the Declaration of Helsinki, and approved by the Institutional Review Board of the University hospital of Tuebingen (approval number 696/2021B01).

**Informed Consent Statement:** Informed consent was obtained from all subjects involved in the study.

**Data Availability Statement:** Not applicable.

**Conflicts of Interest:** F.H. has no conflicts of interest. L.W. has no conflict of interest. J.F. received institutional research support from Siemens Healthcare USA, DePuy, Zimmer, Microsoft, and BTG International, and is a scientific advisor of Siemens Healthcare USA, Alexion Pharmaceuticals, and BTG International. He also received speaker's honorarium from Siemens Healthcare USA and has shared patents with Siemens Healthcare and Johns Hopkins University. R.G. is employee of Siemens Healthcare. B.S. is employee of Siemens Healthcare. S.F. is employee of Siemens Healthcare. F.B. received speaker's honorarium and an unrestricted research grant from Siemens Healthcare. He also received speaker's honorarium and an unrestricted research grant from Bayer Healthcare. He is at the advisory board of Bayer Healthcare. S.S. has no conflict of interest. K.N. has no conflict of interest. M.H.: received institutional research support from Siemens Healthineers Germany; he is a scientific advisor of Siemens Healthineers Germany and received speaker's honorarium from Siemens Healthineers Germany.

## Abbreviation

| | |
|---|---|
| CNR | Contrast-to-Noise Ratio |
| DSCT | Dual-Source CT |
| PCD-CT | Photon-Counting Detector CT |
| SNR | Signal-to-Noise Ratio |
| SSDE | Size-Specific Dose Estimation |

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
