# Peer review of "Image Quality and Radiation Dose of Contrast-Enhanced Chest-CT Acquired on a Clinical Photon-Counting Detector CT vs. Second-Generation Dual-Source CT in an Oncologic Cohort: Preliminary Results"

_tomography, doi:10.3390/tomography8030119_

Round 1

Reviewer 1 Report

This study compares two different CT modalities in oncological diagnostics.

Some comments:

Material and Methods 

Statistics

Is there any power calculation regarding the choice of sample size? On what basis did the authors include 100 subjects?

Subjects 

A clarification about the subjects is of interest. Are the subjects matched pairs? How was the matching procedure performed? According to tumor size, patient characteristics? 

Subjective Image Quality 

Individual window settings and free-scrolling indicates that the areas which were subjectively assessed were not the same for the 2 radiologists. A comment on why this approach, instead of specifically using the same areas to asses, is of interest for the reader.  

Furthermore, a clarification on how the consensus round was performed is of importance. Did the radiologists then sit together to reach an agreement? How many times did this happen in the study? And how did this impact the statistics? 

Results 

Patient Characteristics  

More information on gender differences, creatinine difference and tumor size is of interest. 

Image Quality Quantification 

It might have been of interest also to evaluate tumor characteristics and lung opacities to see if there was a difference. A clarification on why this was not performed is of interest. 

Subjective Image Quality 

As noted previously, a clarification on where the image quality is performed is of interest. Did the radiologists compare the same areas or did the freely choose an area? 

In table 3 there is only listed a median number with a min and max. A table with the individual readers assessment is of interest. How many times did the readers disagree and how many times did a consensus assessment take part? 

Discussion 

Are there other differences in PCD-CT compared do DSCT that favors one over the other? Specifically, is there a difference in time in scan acquisition, post-processing etc?   

In the discussion, the authors refer to previous studies in where image quality is dependent on patient characteristics, tumor size, however this is not addressed in this study. More information regarding patient characteristics, gender, creatinine and tumor size is of interest for the reader both in the results and the discussion part. 

A discussion on how the images were assessed by the radiologists if of interest in the discussion part. How many times did the readers disagree and was there a difference between the PCD-CT and DSCT? 

Author Response

May 14, 2022

RE: Revised manuscript with the title: “Image Quality and Radiation Dose of Contrast-Enhanced Chest CT acquired on a Clinical Photon-Counting Detector CT vs. 2nd Generation Dual-Source CT in an oncologic cohort”

Dear Reviewer,

We are very appreciative to be given the opportunity to revise the manuscript titled “Image Quality and Radiation Dose of Contrast-Enhanced Chest CT acquired on a Clinical Photon-Counting Detector CT vs. 2nd Generation Dual-Source CT in an oncologic cohort” and to resubmit it for your consideration for publication in Tomography.

We would like to thank for the very helpful suggestions and comments to improve our manuscript. We addressed all aspects raised by the reviewers and added comments to these points in the highlight section, the introduction, the M&M section, the results and to the discussion.

This research is original and has not been submitted to any other journals. All authors have approved the manuscript as well as the author changes and agreed on its resubmission to Tomography.

Some comments:

Material and Methods 

Statistics

Is there any power calculation regarding the choice of sample size? On what basis did the authors include 100 subjects?

Due to the consistent differences in image quality as well as radiation dose between the two scanners, we decided to analyze this interims data earlier as initially planed and therefore we added “preliminary results” to the manuscript title.

Subjects 

A clarification about the subjects is of interest. Are the subjects matched pairs? How was the matching procedure performed? According to tumor size, patient characteristics? 

This cohort consisted all the time of the same patients (age, gender, body weight, etc.) who were examined twice at follow-up within three months using first an EID-CT and later a PCD-CT (the PCD-CT was the successor of the EID-CT in our department).

Subjective Image Quality 

Individual window settings and free-scrolling indicates that the areas which were subjectively assessed were not the same for the 2 radiologists. A comment on why this approach, instead of specifically using the same areas to asses, is of interest for the reader.  

Within the region of interest, both radiologists were free to choose their slice level. However, both readers had to complete a questionnaire suggesting the anatomical areas to be evaluated (e.g. lung upper lobes, the pulmonary arteries shortly before their branching, descendent thoracic aorta about 3 cm above the diaphragms) and the ROI ranges that should be generally used. We added this information to our M&M part and hope you might agree with it (see Page 4).

Furthermore, a clarification on how the consensus round was performed is of importance. Did the radiologists then sit together to reach an agreement? How many times did this happen in the study? And how did this impact the statistics? 

The consensus reading consisted of a third joint measurement using a new ROI accepted by both readers. It happens ten time. We added this information to our M&M and results parts:

“Disagreements were resolved during a final consensus round (n=10 cases). The consensus reading consisted of a third joint measurement using a new ROI set together by both readers.” (Page 4).

After the first reading session 10 image data sets (4 male and 6 female patients) were rated different by both radiologists. A common consent was reached after the final reading session.” (Page 9).

Results 

Patient Characteristics  

More information on gender differences, creatinine difference and tumor size is of interest. 

There were no significant changes in serum creatinine levels at follow-up.

Please note that the same patients were examined twice within three months as part of a treatment monitoring schedule!

With regard to the tumor size and image quantification we added the corresponding tumor contrast to Table N°3 and added the tumor size. We hope that you agree with our changes.

“Q1 and Q3 strength resulted in a significantly higher tumor tumor-to to-lung parenchym contrast ratio compared to the corresponding DSCT data sets. The maximum tumor size was significantly larger at the time of PCD acquisition (12.4 ± 8.96 cm vs. 11.00 ± 8.63 cm)” (Page 7)

Moreover, we give supplementary information on the oncological background in table N°1:

Table 1: Distribution of oncoligical disease among the participants

Oncological Diseases

Absolute value (Relative value in %)

NSCLC/SCLC

12 (12%) / 7 (7%)

Colorectal carcinoma

17 (17%)

Gastrooesophageal carcinoma

15 (15%)

Pancreatic carcinoma

10 (10%)

Hepatobilliary cancer

8 (8%)

Lymphoma

7 (7%)

Others (ovarian carcinoma, thymic carcinoma, etc.)

24 (24%)

Image Quality Quantification 

It might have been of interest also to evaluate tumor characteristics and lung opacities to see if there was a difference. A clarification on why this was not performed is of interest. 

We fully agree with you, but this was not the focus of this first study of us. Nevertheless, preliminary reports from the literature suggest a potential benefit in terms of increased spatial resolution (Quantum Plus HR) and also improved lesion delineation due to less image noise.

Nevertheless, at your suggestion we additionally calculated the tumor-to-lung parenchyma ratio in patients with lung metastases emphasizing the benefit of increased tissue contrast (see M&M and Results):

“In patients presenting with pulmonary lesions (all of them metastatic in origin) (n=23), an additional ROI was placed in the main tumor manifestation (ROI size, 150-500mm2) and a ratio to the attenuation of lung parenchyma was calculated according to the formula:

Tumor-to-lung parenchyma contrast ratio = |(ROItumor/ROIlung parenchyma)|” (Page 4)

“Q1 and Q3 strength resulted in a significantly higher tumor-to-lung parenchym contrast ratio compared to the corresponding DSCT data sets. The maximum tumor size was significantly larger at the time of PCD acquisition (12.4 ± 8.96 cm vs. 11.00 ± 8.63 cm) (Table 3 and Figure 1 and 2).” (Page 7).

DSCT

PCD-CT

p-value

corrected p-value

Q1

Q3

Tumor-to-lung parenchyma contrast ratio

(Median ± SD)

0.03 ± 0.02

0.08 ± 0.04

0.08 ± 0.04

<0.001*

**Q3>DSCT (p<0.001)

**Q1>DSCT (p<0.001)

*Friedman-test, **Post-Hoc Dunn-Bonferroni-Tests with corrected alpha

Subjective Image Quality 

As noted previously, a clarification on where the image quality is performed is of interest. Did the radiologists compare the same areas or did the freely choose an area? 

We added for clarification following sentences:

“Within the region of interest, both radiologists were free to choose their slice level. However, both readers had to complete a questionnaire suggesting the anatomical areas to be evaluated (e.g. lung upper lobes, the pulmonary arteries shortly before their branching, descendent thoracic aorta about 3 cm above the diaphragms)” (Page 4).

In table 3 there is only listed a median number with a min and max. A table with the individual readers assessment is of interest. How many times did the readers disagree and how many times did a consensus assessment take part? 

Thank you for your query. We added the information to our M&M and Results parts:

“Disagreements were resolved during a final consensus round (n=106 cases). The consensus reading consisted of a third joint measurement using a new ROI set together by both readers.” (Page 4).

After the first reading session 10 image data sets (4 male and 6 female patients) were rated different by both radiologists. A common consent was reached after the final reading session.” (Page 9).

Discussion 

Are there other differences in PCD-CT compared do DSCT that favors one over the other? Specifically, is there a difference in time in scan acquisition, post-processing etc?

We addressed further potential advantages of PCD over EID both in the Introduction and Discussion section based on literature reports. However, this was not the primarily focus of this preliminary study of us. As we additionally calculated also the tumor-to-lung parenchyma ratio, we added a sentence referring to in the discussion section.

“Some major aspects related to the superiority of PCD over EID in the chest diagnosis in terms of lesion detectability, improved spatial resolution for accurate display and interpretation of e.g. interstitial lung diseases have not been addressed in this study.” (Page 12).

In the discussion, the authors refer to previous studies in where image quality is dependent on patient characteristics, tumor size, however this is not addressed in this study. More information regarding patient characteristics, gender, creatinine and tumor size is of interest for the reader both in the results and the discussion part. 

Thank you for your query. We added the information to our discussion and hope you might agree with it:

Finally, the tumor-to-lung parenchyma contrast ratio proved superior for the PCD over the EID” (Page 11).

A discussion on how the images were assessed by the radiologists if of interest in the discussion part. How many times did the readers disagree and was there a difference between the PCD-CT and DSCT? 

We added this information as requested.

Reviewer 2 Report

Very interesting and well written article comparing two novel CT techniques. The report is original, and the manuscript is overall very well written with good methodology. Introduction should be expanded, to give a more exhaustive overview of the techniques' potential (especially photon counting CT) and introduce the study questions.

In the matherials and methods section incrlusion and exclusion criteria shoul be clarified: was the enrolement prospective or retrospective? were patients submitted to chest CT angiography (an arterial phase is described)? For what indication? How were patients addressed to both techniques? Due to technical clinical needs or per study protocol?

Author Response

May 14, 2022

RE: Revised manuscript with the title: “Image Quality and Radiation Dose of Contrast-Enhanced Chest CT acquired on a Clinical Photon-Counting Detector CT vs. 2nd Generation Dual-Source CT in an oncologic cohort”

Dear Reviewer,

We are very appreciative to be given the opportunity to revise the manuscript titled “Image Quality and Radiation Dose of Contrast-Enhanced Chest CT acquired on a Clinical Photon-Counting Detector CT vs. 2nd Generation Dual-Source CT in an oncologic cohort” and to resubmit it for your consideration for publication in Tomography.

We would like to thank for the very helpful suggestions and comments to improve our manuscript. We addressed all aspects raised by the reviewers and added comments to these points in the highlight section, the introduction, the M&M section, the results and to the discussion.

This research is original and has not been submitted to any other journals. All authors have approved the manuscript as well as the author changes and agreed on its resubmission to Tomography.

Very interesting and well written article comparing two novel CT techniques. The report is original, and the manuscript is overall very well written with good methodology. Introduction should be expanded, to give a more exhaustive overview of the techniques' potential (especially photon counting CT) and introduce the study questions.

We extended the technical description of the two scanners in the Introduction section, as requested.

“As PCDs do not have scintillators and septa they can be fabricated with smaller elements compared to DSCT, thus significantly improving the spatial resolution [13]. Moreover, image quality can be further improved by manipulating the weighting of energy bins in the spectral data and also by using the decomposition of the material. For chest-CT applications, a strong improvement in the detectability of both lesions with low-contrast compared to the adjacent lung parenchyma (e.g. partial solid nodules) as well as for high-contrast lesions has been reported by Si-Mohamed et al. [14]” (Page 2).

In the matherials and methods section inclusion and exclusion criteria should be clarified: was the enrolement prospective or retrospective? were patients submitted to chest CT angiography (an arterial phase is described)? For what indication? How were patients addressed to both techniques? Due to technical clinical needs or per study protocol?

This was a prospective study evaluation as stated in the first sentence of the M&M section.

All patients were enrolled consecutively and none of them had to be excluded from the final evaluation. As differences both in image quality and radiation dose became evident early in the course of the study, we decided to make a preliminary analysis of this cohort.” (Page 3).

All patients were enrolled consecutively and none of them had to be excluded from the final evaluation. As differences both in image quality and radiation dose became evident early in the course of this study, we decided to make a preliminary analysis of our cohort. Therefore, we declared this study as “preliminary” (please see the completed title).

Reviewer 3 Report

very interesting  comparison  well  presented with  good   scientific   back ground  good   choice  of   ROI's   for this  study 

Interesting  for the  readers of  "Tomography"

I  would   like   to  have more   examples  in  article   figures  presenting differences    between  PCD--CT  and   DSCT   on    different  windows  with  different   Kernell  used    -   this   might  be  more  convincing  for the readers  

Is   PCD - CT  is  nor  producing more  artifacts -   this  matter  should   be  presented more   clearly  in the  paper  

Author Response

May 14, 2022

RE: Revised manuscript with the title: “Image Quality and Radiation Dose of Contrast-Enhanced Chest CT acquired on a Clinical Photon-Counting Detector CT vs. 2nd Generation Dual-Source CT in an oncologic cohort”

Dear Reviewer,

We are very appreciative to be given the opportunity to revise the manuscript titled “Image Quality and Radiation Dose of Contrast-Enhanced Chest CT acquired on a Clinical Photon-Counting Detector CT vs. 2nd Generation Dual-Source CT in an oncologic cohort” and to resubmit it for your consideration for publication in Tomography.

We would like to thank for the very helpful suggestions and comments to improve our manuscript. We addressed all aspects raised by the reviewers and added comments to these points in the highlight section, the introduction, the M&M section, the results and to the discussion.

This research is original and has not been submitted to any other journals. All authors have approved the manuscript as well as the author changes and agreed on its resubmission to Tomography.

very interesting  comparison  well  presented with  good   scientific   back ground  good   choice  of   ROI's   for this  study 

Interesting  for the  readers of  "Tomography"

I  would   like   to  have more   examples  in  article   figures  presenting differences    between  PCD--CT  and   DSCT   on    different  windows  with  different   Kernell  used    -   this   might  be  more  convincing  for the readers  

At your request we have introduced some more figures comparing EID with PCD scanner. (See figure 1 and 5).

Is   PCD - CT  is  nor  producing more  artifacts -   this  matter  should   be  presented more   clearly  in the  paper  

Thank you for your query. We added the following sentence to our discussion and hope you might agree with it:

“Regarding artifacts, PCD-CT also has the potential to reduce blooming by means of improved spatial resolution and material decomposition [19].” (Page 12).

Reviewer 4 Report

The manuscript is well written, and the study is well-conducted and presented. The topic is overage regarding its appeal to readers but the authors should be certainly commended for their efforts in doing such a good job. I recommend a double-check for typos and a light English grammar and structure revision.

Author Response

May 14, 2022

RE: Revised manuscript with the title: “Image Quality and Radiation Dose of Contrast-Enhanced Chest CT acquired on a Clinical Photon-Counting Detector CT vs. 2nd Generation Dual-Source CT in an oncologic cohort”

Dear Reviewer,

We are very appreciative to be given the opportunity to revise the manuscript titled “Image Quality and Radiation Dose of Contrast-Enhanced Chest CT acquired on a Clinical Photon-Counting Detector CT vs. 2nd Generation Dual-Source CT in an oncologic cohort” and to resubmit it for your consideration for publication in Tomography.

We would like to thank for the very helpful suggestions and comments to improve our manuscript. We addressed all aspects raised by the reviewers and added comments to these points in the highlight section, the introduction, the M&M section, the results and to the discussion.

This research is original and has not been submitted to any other journals. All authors have approved the manuscript as well as the author changes and agreed on its resubmission to Tomography.

The manuscript is well written, and the study is well-conducted and presented. The topic is overage regarding its appeal to readers but the authors should be certainly commended for their efforts in doing such a good job. I recommend a double-check for typos and a light English grammar and structure revision.

Thank you for your great feedback. We double checked the manuscript for typos, grammar and structure. We hope you might agree with our changes.

Round 2

Reviewer 1 Report

The manuscript has been satisfactory revised.

Reviewer 2 Report

Minor comments correctly addressed